# Original speech and its echo are segregated and separately processed in the human brain

**Jiaxin Gao[1], Honghua Chen[1], Mingxuan Fang[1], Nai Ding**[1,2,3]*

**1** Key Laboratory for Biomedical Engineering of Ministry of Education, College of Biomedical Engineering and Instrument Sciences, Zhejiang University, Hangzhou, China, **2** Nanhu Brain-computer Interface Institute, Hangzhou, China, **3** The State key Lab of Brain-Machine Intelligence; The MOE Frontier Science Center for Brain Science & Brain-machine Integration, Zhejiang University, Hangzhou, China

* ding_nai@zju.edu.cn

## Abstract

Speech recognition crucially relies on slow temporal modulations (<16 Hz) in speech. Recent studies, however, have demonstrated that the long-delay echoes, which are common during online conferencing, can eliminate crucial temporal modulations in speech but do not affect speech intelligibility. Here, we investigated the underlying neural mechanisms. MEG experiments demonstrated that cortical activity can effectively track the temporal modulations eliminated by an echo, which cannot be fully explained by basic neural adaptation mechanisms. Furthermore, cortical responses to echoic speech can be better explained by a model that segregates speech from its echo than by a model that encodes echoic speech as a whole. The speech segregation effect was observed even when attention was diverted but would disappear when segregation cues, i.e., speech fine structure, were removed. These results strongly suggested that, through mechanisms such as stream segregation, the auditory system can build an echo-insensitive representation of speech envelope, which can support reliable speech recognition.

## Introduction

Extracting auditory objects reliably from a complex auditory scene is a primary goal of auditory processing [1–4]. For humans, speech is the most important communicational sound and humans can reliably recognize speech in various listening environments [5,6]. Previous studies have suggested that the neural representation of speech envelope, i.e., the fluctuation in speech power over time [7,8], is resilient to many types of acoustic degradation at the level of auditory cortex [9–16], showing that distinct neural mechanisms are required to compensate for different types of acoustic degradation. For example, stationary noise can greatly attenuate the dynamic range of temporal modulations [17], which is an effect that can be well compensated by relatively basic neural adaptation mechanisms [9,10,18]. When the speech is mixed with a competing voice, however, neural adaptation to the overall statistical properties of the sound mixture can barely help to enhance the neural response to the target speech stream. Instead, the brain has to segregate the sound mixture into auditory streams by analyzing, e.g., pitch and spatial cues, and selectively process the auditory stream of interest under the modulation of

**Data Availability Statement:** The data and analysis codes that can generate the figures are available at https://zenodo.org/records/10472483 (DOI: 10.5281/zenodo.10472483).

**Funding:** This work was supported by the National Natural Science Foundation of China (https://www.nsfc.gov.cn/) (32222035 to ND) and Key R & D Program of Zhejiang (https://kjt.zj.gov.cn/) (2022C03011 to ND). The funders had no role in study design, data collection and analysis, decision to publish, or preparation of the manuscript.

**Competing interests:** The authors have declared that no competing interests exist.

**Abbreviations:** DFT, discrete Fourier transformation; FDR, false discovery rate; FIR, finite impulse response; tSSS, temporal signal space separation; TRF, temporal response function.

top-down attention [2–4,19]. In such conditions, if speech segregation cues are removed through, e.g., noise vocoding, speech segregation fails and the speech recognition rate drops [20,21].

A large number of studies have focused on the neural encoding of speech envelope since the speech envelope provides crucial cues for speech recognition [22,23]. When the speech envelope is further decomposed into faster and slower components, referred to as temporal modulations [24,25], it has been demonstrated that temporal modulations between 1 and 8 Hz are essential for speech recognition: When the temporal modulations between 1 and 8 Hz are removed, a large number of studies consistently reported that speech intelligibility drops [26–29]. Recently, however, it has been demonstrated that a single echo can eliminate the temporal modulations at frequencies that are determined by the echo delay. For example, an echo with a 125-ms delay can eliminate temporal modulations at 4 Hz and is supposed to reduce speech intelligibility (Fig 1B). Nevertheless, young listeners show no difficulty recognizing such echoic speech [30]. The high intelligibility of echoic speech indicates that either the temporal modulations eliminated by an echo are neurally restored or that these temporal modulations are actually not necessary for speech recognition, in contradiction to the conclusions of a large number of previous studies [26–29].

Echoes can be viewed as a special case for reverberation since both echoes and reverberation are generated by sound reflections. For a single echo, the power of sound reflection concentrates at a single delay. In daily reverberant environments, however, the power of sound reflection decays exponentially as the delay of the reflection increases [31]. Echoic environments are prevalent during online conferencing [32]. Previous studies have shown that the influence of reverberation on speech envelope can be compensated through basic neural adaptation mechanisms [9,33]. Furthermore, evidence has been provided that a click and its echo would be encoded and perceived as separate auditory objects [34]. It remains unclear, however, whether these mechanisms can restore the temporal modulations eliminated by echoes.

The current study investigated the neural encoding of echoic speech using MEG and probed the underlying neural mechanisms through computational models. We added a single echo to a narrated story (Fig 1A) to create a strong influence on temporal modulations that are important to speech recognition. In such challenging echoic environments, we used MEG to investigate whether the auditory cortex can track the temporal modulations that are attenuated or eliminated by an echo. If neural activity can track these missing components, we further hypothesized that the underlying neural mechanisms may include neural adaptation or auditory stream segregation. The neural adaptation hypothesis was tested using a computational model of synaptic depression and gain control, while the stream-segregation hypothesis was tested by analyzing whether speech and its echo have distinct spatiotemporal representations using the temporal response function (TRF) models.

## Results

### Echo-related suppression of temporal modulations

The echoic speech was a mixture of 2 speech signals that only differed by a time delay (Fig 1A). In the following, the leading sound was referred to as the direct sound, and the lagging sound was referred to as the echo. We first constructed a challenging echoic condition in which the echo had the same amplitude as the direct sound. The echo delay was either 0.125 s or 0.25 s (see Methods for rationales). A modulation spectrum analysis clearly revealed that an echo could notch out the temporal modulations at some frequencies that were related to the echo delay (Fig 1B and 1C). Fig 1A illustrates why these notches were created—the Fourier analysis decomposed a signal into sinusoids. If the signal was time shifted by $T$, forming an echo that

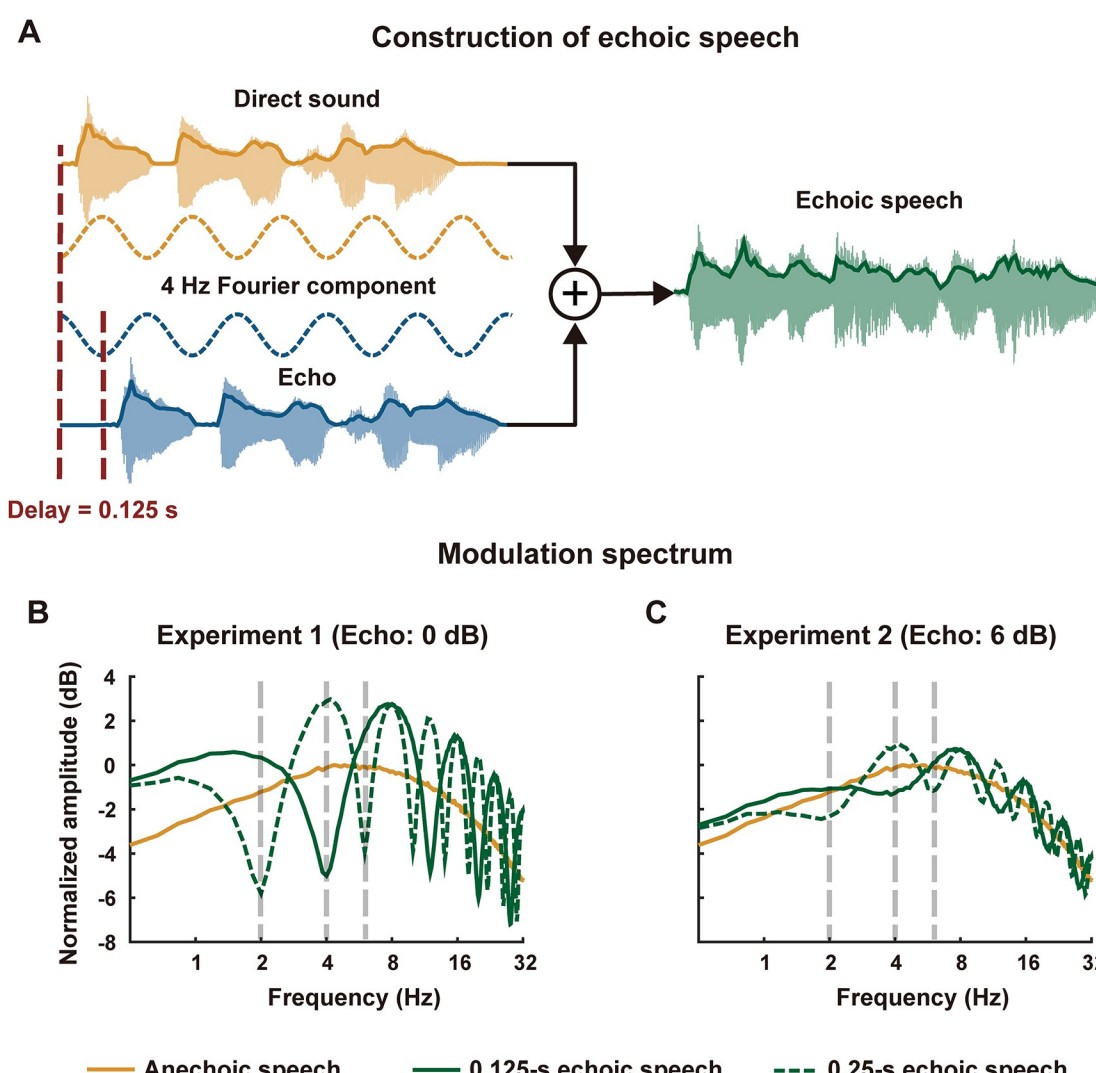

**Fig 1. Construction of echoic speech and the modulation spectrum. (A)** In Experiment 1, the echoic speech is generated by delaying a speech signal by $T$ (0.125 s or 0.25 s) and adding together the delayed signal with the original signal. The dashed sinusoids in yellow and blue represent the $1/2T$ Hz Fourier components of the envelope of direct sound and echo, respectively. The 2 sinusoids are 180 degrees out of phase and are fully canceled in the mixture. **(B)** Modulation spectrum of stimulus in Experiment 1. Dashed gray lines denote the echo-related frequencies below 10 Hz. **(C)** Modulation spectrum of stimulus in Experiment 2, in which the amplitude of the delayed signal is twice the amplitude of the direct sound (6-dB echo). The underlying data can be found at https://zenodo.org/records/10472483.

had the same amplitude as the direct sound, all its Fourier components would also be shifted by $T$. Consider a sinusoidal Fourier component whose period was $2T$, which had an opposite phase for the original signal and the echo, would get canceled when the original signal and the echo were mixed. The same applied to Fourier components whose periods were $2T/3$, $2T/5$, etc. In the following, the frequencies of these sinusoidal components that were notched out by the echo, i.e., $1/2T$, $3/2T$, $5/2T$, were referred to as the echo-related frequencies. Furthermore, since previous studies showed that the cortical responses mainly track the speech envelope below 10 Hz [35], we only analyzed the echo-related frequencies below 10 Hz.

The power spectrum analysis could effectively reveal how the speech envelope was influenced by an echo, but it could not be directly applied to quantify the envelope-tracking neural

response in the MEG recording, since the power of MEG response is dominated by, e.g., spontaneous neural activity. To isolate the envelope-tracking response, we analyzed the phase coherence between the MEG response and the temporal envelope of direct sound [36–38]. The phase coherence spectrum quantified the phase locking between 2 signals in different frequencies. Specifically, the response phase of each signal was extracted in consecutive time windows using the Fourier transform. If the 2 signals were perfectly synchronized, the phase lag between them would be a constant across all time windows and the phase coherence would reach its maximum, i.e., 1. In contrast, if the 2 signals were independent of each other, their phase lag would be random and the phase coherence would be at a chance level. Our hypothesis, referred to as the envelope restoration hypothesis, was that the auditory system could fully or partially restore the temporal envelope of direct sound, and an alternative hypothesis was that the auditory system faithfully followed the temporal envelope of the echoic speech. We first quantified the prediction of the alternative hypothesis through a simulation, in which the neural response was simply simulated using the envelope of echoic speech. The phase coherence spectrum between the simulated neural response and the envelope of direct sound showed notches at 4 and 12 Hz for speech with a 0.125-s echo and showed notches at 2, 6, 10, and 14 Hz for speech with a 0.25-s echo (Fig 2A). These results demonstrated that if neural activity faithfully tracked the envelope of echoic speech, it would show very low phase coherence (<0.07) with the envelope of the direct sound at echo-related frequencies. Therefore, if the phase coherence between the neural response to echoic speech and the temporal envelope of direct sound is near 0 at echo-related frequencies, the alternative hypothesis is supported. Otherwise, the envelope restoration hypothesis is supported and full restoration is suggested if the phase coherence value is comparable to the neural responses to anechoic speech and direct sound.

In Experiment 1, the participants listened to the anechoic speech, 0.125-s echoic speech, and 0.25-s echoic speech in separate blocks (S1 Audio), while the cortical responses were recorded using MEG. In the anechoic speech condition, listeners only listened to the direct sound without being accompanied by an echo. In the echoic speech conditions, the magnitude of the echo was the same as the direct sound (0-dB echo). Participants were asked to attend to speech and answer comprehension questions after each block. The average accuracy for question-answering was 95.6% for the 3 conditions. We characterized the neural tracking of the direct sound using phase coherence spectrum between the MEG responses and the temporal envelope of the direct sound, which was in the same way as the simulation analysis in Fig 2A. The phase coherence spectrum was calculated for each gradiometer and the average overall gradiometers was shown in Fig 2B. In the anechoic condition, i.e., when the stimulus only included the direct sound, the phase coherence between the MEG response and speech envelope was significantly above chance below approximately 10 Hz (Fig 2B, $p < 0.05$, permutation test, false discovery rate (FDR) corrected). Within this frequency range, i.e., below 10 Hz, we would probe whether the MEG activity can track the speech envelope at echo-related frequencies when the participants listening to echoic speech.

For the echoic conditions, the phase coherence was also significantly above chance at the echo-related frequencies (Fig 2B and 2C, $p < 0.05$ for all echo-related frequencies, permutation test, FDR corrected). The response topography at the echo-related frequency revealed bilateral temporal distribution (Fig 2D). Compared with the anechoic condition, the 0.125-s echoic condition did not show a significant reduction in phase coherence at the echo-related frequency, e.g., 4 Hz, (Fig 2B, left plot, and 2C, $p = 0.4235$ at 4 Hz, permutation test, FDR corrected) and the 0.25-s echoic condition did not show a significant reduction in phase coherence at 2 Hz (Fig 2B, right plot, and 2C, $p = 0.3941$ at 2 Hz, permutation test, FDR corrected), suggesting full restoration. At 6 Hz, however, the phase coherence in the 0.25-s echoic condition is significantly above chance but lower than the phase coherence in the anechoic

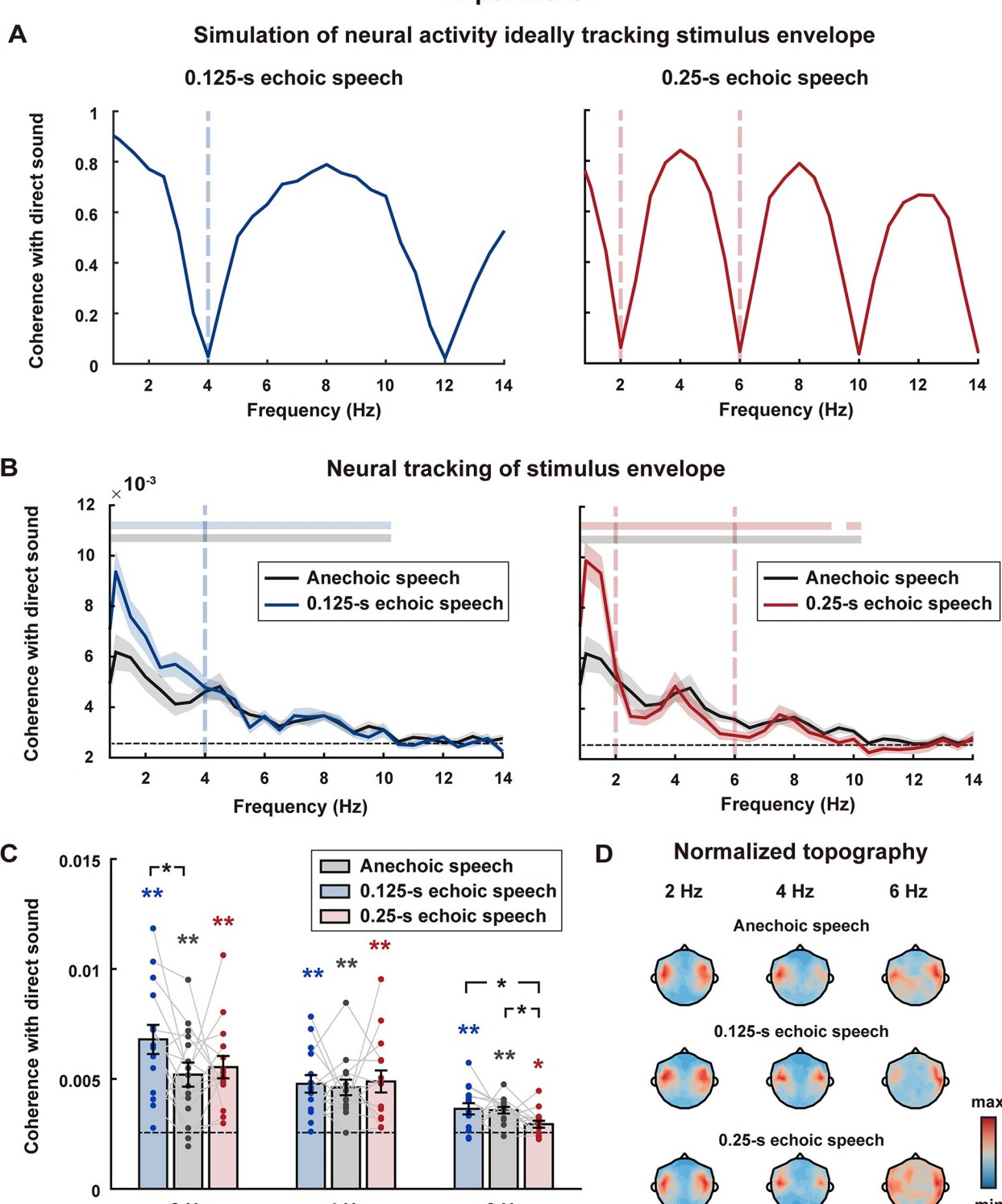

**Fig 2. Results of Experiment 1.** (**A**) Simulation of neural activity ideally tracking the envelope of echoic speech. (**B**) Phase coherence spectrum between the MEG response and the envelope of direct sound, averaged over participants and MEG gradiometers. Shaded areas cover 1 SEM across participants on each side. The dashed black line near the bottom shows chance-level phase coherence. Bars on top denote the frequency bins with significant phase coherence ($p < 0.05$, permutation test, FDR corrected). (**C**) Phase coherence at echo-related frequencies. Each dot represents 1 individual and error bars represent 1 SEM. Dashed black lines show chance-level phase coherence. Phase coherence significantly higher than

chance level and significant differences between conditions are marked (* $p < 0.05$, ** $p < 0.01$, permutation test, FDR corrected). **(D)** Topography of gradiometers for the phase coherence at echo-related frequencies, normalized by dividing its maximum. The underlying data can be found at https://zenodo.org/records/10472483.

condition (Fig 2B, right plot, and 2C, $p = 0.0168$, permutation test, FDR corrected), suggesting partial restoration. Furthermore, the phase coherence around 1 Hz was enhanced in the echoic conditions and this difference was statistically significant (S2 Fig, $p < 0.001$ in 0.125-s, and $p = 0.0016$ in 0.25-s echoic conditions, permutation test, FDR corrected), which can possibly be explained by the enhancement of 1-Hz modulation power for echoic speech. Taken together, these results suggested that the human auditory system could effectively restore the speech envelope at the echo-related frequencies.

Experiment 1 showed that the temporal envelope of direct sound could be neurally restored when listening to echoic speech, while the underlying mechanisms remain to be explored. One potential mechanism that might contribute to echo suppression is neural adaptation. To test whether neural adaptation was sufficient to explain the MEG response to the echoic speech, we simulated the adapted neural response [9] and calculated the phase coherence between the simulated neural response and the envelope of direct sound (Figs 3B and S1). Simulation showed that the phase coherence at echo-related frequencies was boosted by neural adaptation for echoic speech, but it remained lower for echoic speech than anechoic speech (Fig 3B), suggesting that neural adaptation could partially restore the envelope of direct sound. Therefore, neural adaptation could possibly explain the neural response at 6 Hz when listening to 0.25-s echoic speech, but could not fully explain the neural responses at 2 and 4 Hz for the 0.25-s and 0.125-s echoic speech conditions.

When the echo and the direct sound have equal amplitude, their Fourier components at echo-related frequencies are fully canceled. When the echo amplitude deviates from the amplitude of the direct sound, it is less effective at canceling the speech envelope at echo-related frequencies. Neural adaptation breaks the balance between the neural responses to the echo and the direct sound by attenuating the echo response, and therefore recovers the temporal envelope at echo-related frequencies. Attenuating the neural response to the echo, however, did not always have a positive effect on restoring the envelope of direct sound. For example, if the echo was stronger than the direct sound (Fig 3A), attenuating the echo response could make the neural responses to echo and the direct sound had a more similar amplitude, canceling the temporal envelope at echo-related frequencies. Motivated by this idea, we constructed Experiment 2, in which the echo was more intense than the direct sound (S2 Audio), to further probe whether neural restoration of speech envelope could be well explained by neural adaptation. If neural adaptation played a dominant role in restoring the speech envelope, the phase coherence between the neural response to echoic speech and the temporal envelope of direct sound should significantly reduce at echo-related frequencies compared with Experiment 1.

Experiment 2 was the same as Experiment 1 except that the echo was 6 dB more intense than the direct sound. Simulations showed that the neural adaptation aggravated, instead of alleviating, the loss of speech envelope at echo-related frequencies in Experiment 2 (Fig 3B). Participants correctly answered 94.4%, 97.2%, and 100% questions in the anechoic, 0.125-s echoic and 0.25-s echoic conditions, suggesting that they had no troubling understanding the echoic speech even if the echo is more intense than the direct sound. Compared with Experiment 1, the phase coherence in Experiment 2 was lower at 4 Hz ($p = 0.003$, bootstrap, FDR corrected) in the 0.125-s echoic condition and was also lower at 6 Hz in the 0.25-s echoic condition ($p = 0.01$, bootstrap, FDR corrected). No significant difference across Experiments 1 and 2 was observed at 2 Hz in the 0.25-s echoic condition ($p = 0.8031$, bootstrap, FDR

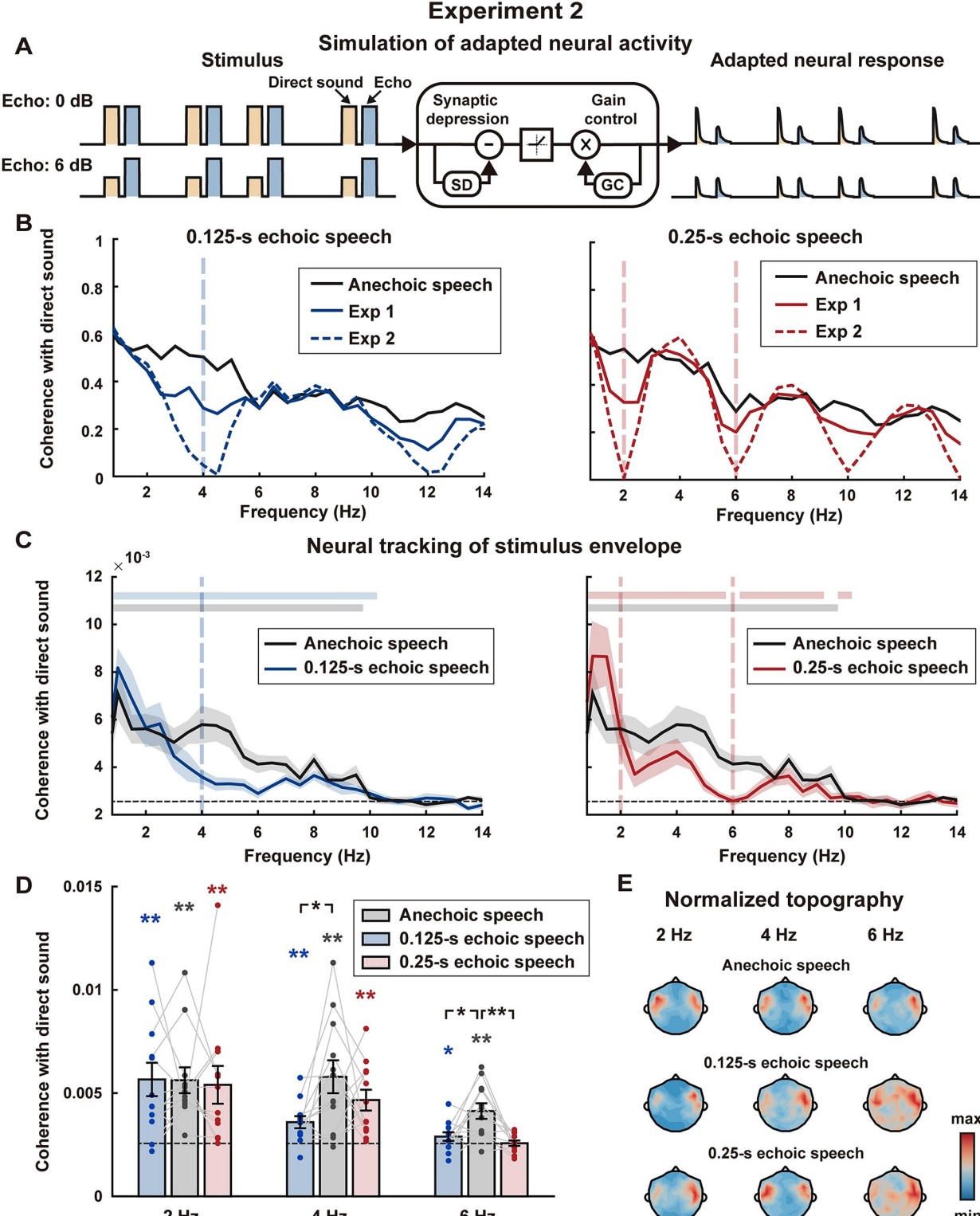

**Fig 3. Results of Experiment 2. (A)** Illustration of neural adaptation. The left part shows the echoic stimulus when the echo was 0 dB and 6 dB, the pulse with the blue background is the direct sound, while the pulse with the yellow background is the echo. The right part shows the adapted neural response simulated using a computational model (Mesgarani and colleagues). **(B)** Phase coherence spectrum between the simulated adapted neural response to echoic speech and the envelope of direct sound. Simulation based on another 2 models is shown in S1 Fig. **(C–E)** The MEG results of Experiment 2 are shown with the same conventions in Fig 2B–2D. The underlying data can be found at https://zenodo.org/records/10472483.

corrected). Additionally, the phase coherence between the MEG response and the envelope of direct sound showed that, for 4 Hz in 0.125-s echoic condition, the phase coherence showed a significant reduction compared to anechoic condition (Fig 3C, left plot, and 3D, $p = 0.0176$, bootstrap, FDR corrected), but was still above chance level (Fig 3C, left plot, and 3D, $p < 0.001$, permutation test, FDR corrected), suggesting partial restoration in the 0.125-s echoic condition. For 2 Hz in the 0.25-s echoic condition, the phase coherence was above chance level at 2 Hz (Fig 3C, right plot, and 3D, $p < 0.001$, permutation, FDR corrected) and did not show a significant reduction compared to the anechoic condition at 2 Hz (Fig 3C, right plot, and 3D, $p = 0.4781$, permutation test, FDR corrected), suggesting full restoration. For 6 Hz in the 0.25-s echoic condition, the phase coherence was not significantly higher than the chance level (Fig 3C, right plot, and 3D, $p = 0.4781$, permutation test, FDR corrected), which was similar to the simulation of neural adaptation. Taken together, consistent with Experiment 1, neural adaptation could possibly explain the neural response in Experiment 2 at 6 Hz, but could not fully explain the neural response at 2 and 4 Hz, suggesting that additional mechanisms are required to restore the speech envelope in echoic conditions.

## Neural segregation of direct sound and echo

Next, we tested whether auditory stream segregation might be involved in the processing of echoic speech. If the echoic speech was neurally segregated into a direct sound and an echo, the 2 streams could be encoded by distinct spatiotemporal neural codes and described by a two-stream TRF model, i.e., the streaming model [12,39] (Fig 4A). We also considered 2 alternative TRF models. The mixture model modeled the MEG response using the temporal envelope of echoic speech, and the idealized model modeled the MEG response using the temporal envelope of direct sound (Fig 4A). We tested whether the streaming model could better explain the MEG response than the other 2 models.

In each echoic condition, the direct sound and echo were fully correlated except for a time delay, making it impossible to separate their neural responses using the TRF model (see Methods for rationales). Therefore, to dissociate the neural responses to the direct sound and the echo, we pooled stimulus conditions with different echo delays (i.e., 0.125-s and 0.25-s delay) in the TRF analysis. The correlation coefficient between the actual MEG response and the response predicted by a TRF model was referred to as the predictive power. The predictive power averaged over all MEG gradiometers was shown in Fig 4B and 4C, left plot. In both experiments, the streaming model had higher predictive power than the mixture model (Fig 4B and 4C, left plot, Experiment 1: $p = 0.0019$; Experiment 2: $p < 0.001$, permutation test, FDR corrected) and the idealized model (Fig 4B and 4C, left plot, $p < 0.001$ for Experiments 1 and 2, permutation test, FDR corrected). When individual MEG channel was considered, the streaming model outperformed the mixture model mostly in the left hemisphere channels (Fig 4B and 4C, middle plot). The TRFs for the 2 streams were illustrated (Fig 4B and 4C, right plot). The TRF for the echo had higher amplitude than the TRF for the direct sound in Experiment 1 but the pattern was reversed in Experiment 2. This result suggested that although the gain of the echo was increased in the stimulus of Experiment 2, the neural response gain for the echo was reduced.

The fact that the streaming model outperformed the 2 alternative models suggested that the auditory system segregated the direct sound and the echo into separate streams and encoded them differently. Next, we constructed additional conditions to further validate the conclusion. Firstly, to further reduce the correlation between the direct sound and the echo, we constructed a variable-delay echoic speech condition, in which the echo delay changed about every 3 s between 0.125 s and 0.25 s. Secondly, speech segregation was enabled by speech

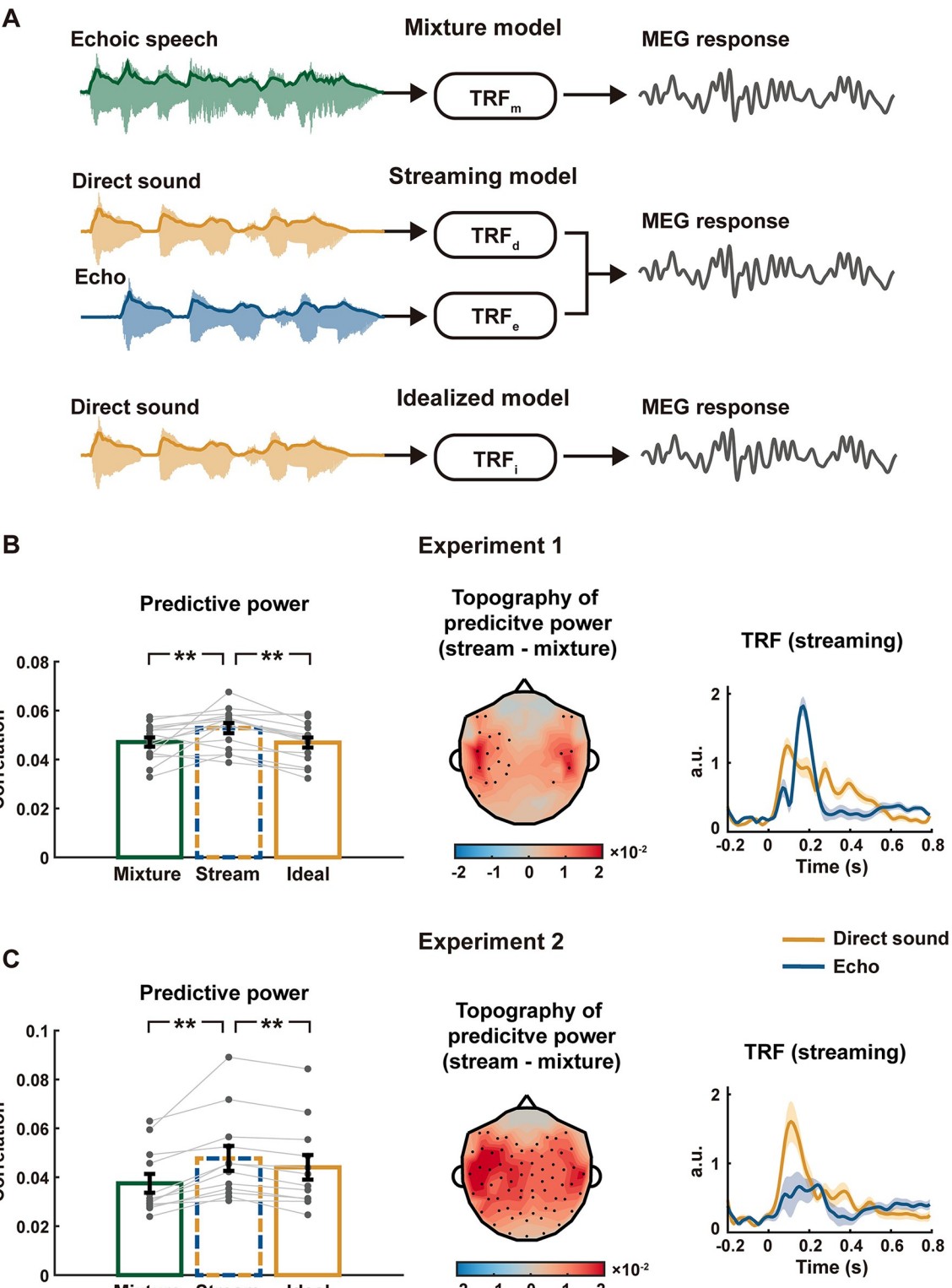

**Fig 4. TRF model and results for Experiments 1 and 2. (A)** Illustration of 3 TRF models. The mixture model only considers the envelope of echoic speech, while the streaming model decomposes echoic speech into a direct sound and an echo and separately models their responses. The idealized model only considers the envelope of the direct sound. **(B, C)** Results of Experiments 1 and 2. **Left**: Bars show the predictive power averaged over participants and MEG gradiometers. Gray dots show individual participants and error bars represent 1 SEM across participants. **Middle**: Topography of gradiometers for the difference between the predictive power of

streaming and mixture models. Black dots indicate sensor locations showing a significant difference between models ($p < 0.05$, bootstrap, FDR corrected). **Right**: TRF for the streaming model (averaged over participants and MEG gradiometers). Shaded areas cover 1 SEM across participants on each side. The underlying data can be found at https://zenodo.org/records/10472483.

segregation cues [3]. Therefore, if we removed the primary speech segregation cues, i.e., the spectro-temporal fine structure [20,21], neural segregation of speech should fail, but neural adaptation mechanisms should not be influenced. Here, we removed the fine structure through 1-channel noise vocoding, which rendered speech unintelligible. Finally, we also manipulated the task to test whether top-down attention was required to segregate the echo from direct sound.

Experiment 3 consisted of 3 conditions. In the first condition, the participants attended to variable-delay echoic speech (S3 Audio) and answered comprehension questions (accuracy = 95.2%). The second condition was the same as the first condition except that the participants watched a silent movie and were asked to ignore the echoic speech. In both conditions, the streaming model had a higher predictive power than the mixture model (Fig 5A and 5B, left plot, $p = 0.0063$ and $p = 0.0046$ for the first and second conditions, respectively, permutation test, FDR corrected) and the idealized model (Fig 5A and 5B, left plot, $p < 0.001$ for the first and second conditions, permutation test, FDR corrected). The advantage of the streaming model lateralized to the left hemisphere channels (Fig 5A and 5B, middle plot). The shape of TRFs in these conditions was similar to that in Experiment 1, in which the direct sound and the echo also had equal amplitude. The third condition presented 1-channel noise vocoded echoic speech (S3 Audio) and the participants watched a silent movie while listening. In this condition, the mixture model better explained the MEG response than the streaming model (Fig 5C, left plot, $p < 0.001$, permutation test, FDR corrected) and the idealized model (Fig 5C, left plot, $p < 0.001$, permutation test, FDR corrected), suggesting that the stimulus was encoded as a whole instead of 2 separate streams.

Additionally, since the phase coherence analysis (Figs 2 and 3) suggested that the higher-frequency responses could be largely explained based on the neural adaptation model, we also separately applied the TRF modeling in 2 frequency bands, i.e., 0.8 to 5 Hz and 5 to 10 Hz. Consistent with the phase coherence analysis, the streaming model outperformed the baseline and idealized models in the lower frequency band, while the mixture model reached the best performance in the higher frequency band (S3 Fig).

Additionally, we could also extend the classic linear TRF model by considering neural adaptation (Figs 6A and S4). The TRF considering neural adaptation significantly outperformed the TRF without neural adaptation in terms of its predictive power (S5 Fig). The relationship between the 3 models, however, remained after considering neural adaptation (Fig 6B–6F). For Experiments 1 and 2, and the first and second conditions of Experiment 3, the adapted streaming model still had the higher predictive power than adapted mixture model (Experiment 1, $p = 0.0019$, Experiment 2, $p < 0.001$, attending to echoic speech, $p = 0.0096$, movie watching (echoic speech), $p = 0.0052$, permutation test, FDR corrected) and adapted idealized model ($p < 0.001$ for Experiment 1, Experiment 2, and first and second conditions in Experiment 3, permutation test, FDR corrected). For the third condition in Experiment 3, the adapted mixture model still better explained the MEG response than the adapted streaming model ($p < 0.001$, permutation test, FDR corrected) and adapted idealized model ($p < 0.001$, permutation test, FDR corrected).

## Discussion

An intense echo strongly distorts speech envelope that contains critical cues for speech intelligibility, but human listeners can still reliably recognize echoic speech. The current study

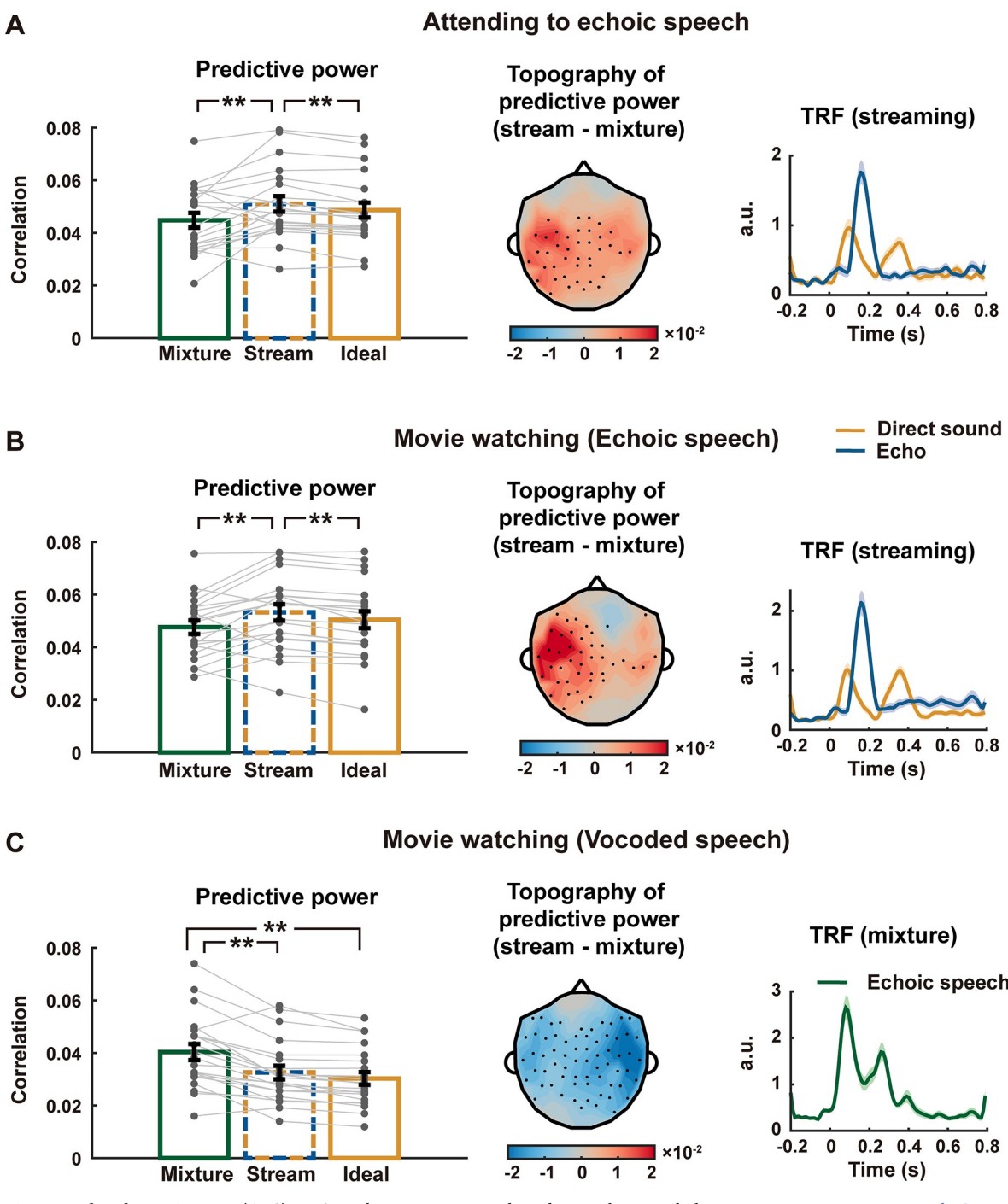

**Fig 5. Results of Experiment 3. (A–C)** MEG results in 3 experimental conditions, shown with the same conventions in Fig 4B and 4C. In panel C, the TRF is shown for the mixture model instead of the streaming model, since the mixture model achieves higher predictive power. The underlying data can be found at https://zenodo.org/records/10472483.

showed that the auditory system can effectively restore the low-frequency components of speech envelope that are attenuated or eliminated by an echo, providing a plausible neural basis for reliable speech recognition. Critically, for the conditions tested in the current study, the echo-related influence on speech envelope cannot be effectively compensated by basic

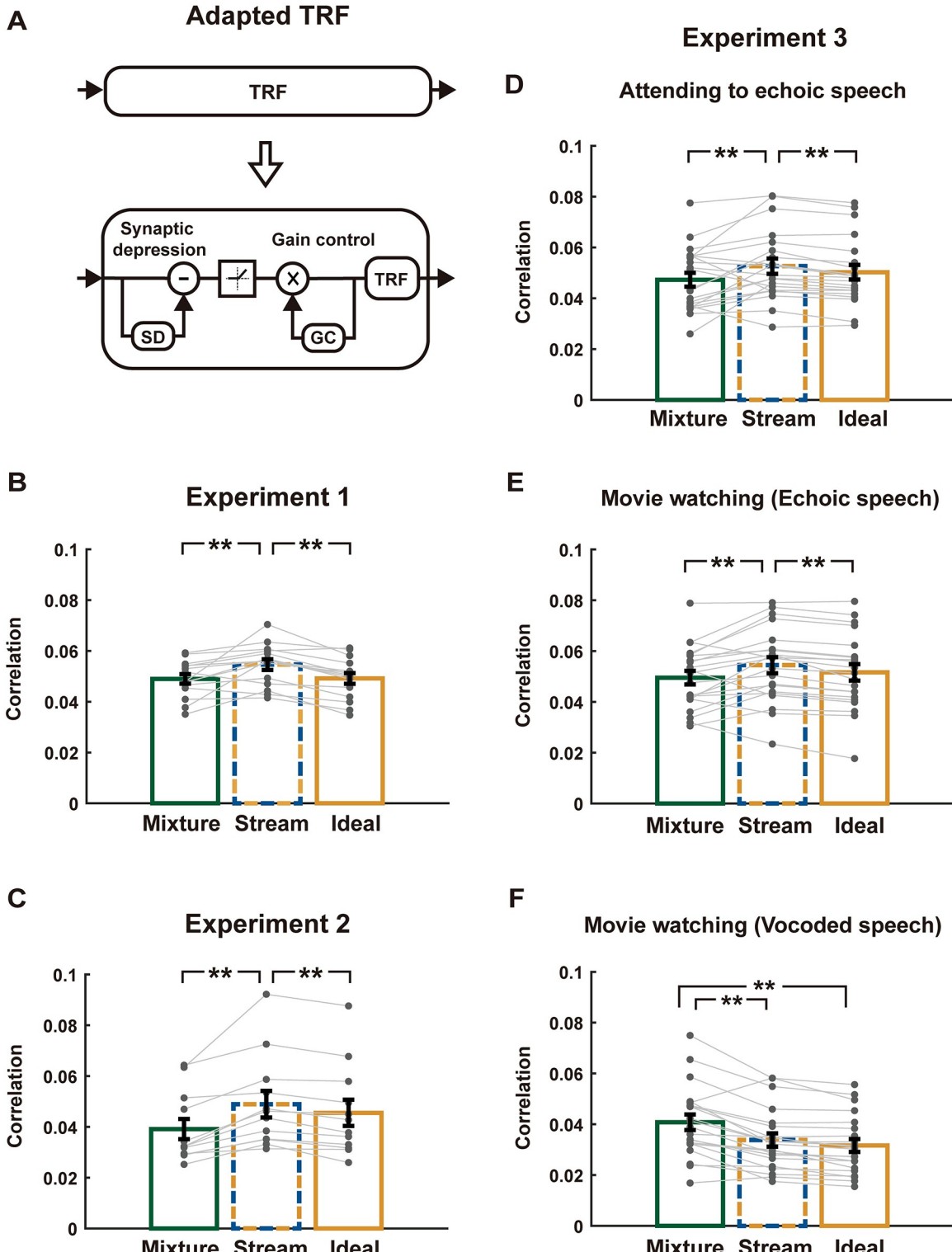

**Fig 6. TRF model based on simulated adapted neural input.** (A) Illustration of the TRF model with adapted neural input—the input to each TRF is adapted based on a computational model (Mesgarani and colleagues). (**B–F**) Predictive powers of the TRF models with adapted neural input, averaged over participants and MEG gradiometers. Gray dots show individual participants. Error bars represent 1 SEM across participants. The underlying data can be found at https://zenodo.org/records/10472483.

neural adaptation mechanisms (Fig 3). Instead, the results strongly suggested that the auditory system can segregate and build separate neural representations for the direct sound and the echo (Figs 4–6). Here, the key difference between the auditory stream segregation and lower-level neural adaptation mechanisms was that auditory stream segregation allowed differential processing of features belonging to the direct sound and the echo. These results showed that auditory stream segregation was not just a key mechanism to achieve robust speech representation in multi-talker environments but might also play a critical role in separating repeated speech streams from the same talker.

Ecologically, an echo can be produced when a distant hard surface generates intense acoustic reflection with >50-ms delay [40]. Reflections with shorter delay are perceptually integrated with the direct sound and can benefit, instead of harming, speech recognition in noisy environments [41]. Echoes are perceptually salient acoustic phenomena. For example, it has been suggested that ancient rock art is often created at places that can generate maximal echo intensities [42], and there are a number of places that are famous for generating echoes, such as the echo wall at the Temple of Heaven in Beijing and the whispering gallery of St Paul's Cathedral in London. During modern teleconferencing, echoes are frequently encountered: An echo is generated when the voice from one side of the conversation is picked up by the microphone on another side and transmitted back [32]. In real teleconference recordings, echoes almost always have >100 ms latency [43].

Neural adaptation is prevalent in the auditory system and provides an obvious candidate to suppress the neural response to echoes, and previous studies have shown that neural adaptation to sound statistics can well suppress the influence of reverberation on speech encoding [9,33]. Furthermore, when 2 brief sounds are presented within a few seconds, neural adaptation could attenuate the response to the second sound [44]. Attenuating the echo to a long-lasting dynamic sound, such as connected speech, however, is much more challenging for at least 2 reasons. Firstly, identifying which sound is an echo is a nontrivial question. For a brief sound, an echo can be defined as a lagging sound. In connected speech, however, a speech segment always follows another speech segment even without any echo, and therefore it is challenging to distinguish which segment belongs to the direct sound and which belongs to the echo. Secondly, since the direct sound and the echo overlap in time and frequency, a simple adaptation strategy cannot selectively suppress the echo while preserving the direct sound. Therefore, although neural adaptation can help suppress echoes to some extent, it cannot fully explain the robust cortical response to echoic speech. Furthermore, when the echo is more intense than the direct sound, synaptic depression tends to aggravate the effect of the echo instead of canceling it (Fig 3A and 3B).

In complex listening environments, such as multi-speaker environments, auditory stream segregation is the basis for robust speech recognition [3,4]. In such environments, the sounds produced by different speakers generally differ in their pitch and spatial locations, which serve as bottom-up cues to segregate speech streams. When 2 utterances of the same speaker are mixed, they cannot be segregated based on bottom-up sensory features [5]. Nevertheless, when prior information is available, e.g., when part of the speech content is known, the brain can also use the prior information to separate speech streams, an effect referred to as schema-based speech segregation [2,45], and this effect can modulate envelope-tracking neural activity [46]. For echoic speech, it is possible that the direct sound serves as prior knowledge to identify the echo. Furthermore, since speech is dynamic, the direct sound and the echo could differ in pitch and spectral modulations at each moment, which provide cues to segregate the direct sound and echo into different streams based on mechanisms such as segregation based on harmonic structure [47], spectro-temporal masking [19], and feature grouping based on temporal coherence [3]. The fact that neural evidence for speech-echo segregation disappears further

demonstrates the importance of pitch and other spectro-temporal fine structure cues in speech-echo segregation. In natural listening environments, the direct sound and the echo differ in their source locations, which can provide additional cues for speech segregation.

Here, the results suggest that neural activity separately tracks the temporal envelope of the direct sound and the temporal envelope of the echo. The temporal envelope of a sound, however, is extracted even in the auditory periphery, while auditory stream segregation occurs in much later auditory processing stages [3,16,24,48,49], leading to an apparent paradox of why auditory streaming can modulate envelope-tracking activity. A solution to the paradox is the following: What is available in the auditory periphery is the temporal envelope of the mixture, while the temporal envelope of the direct sound or the echo can only be resolved after auditory stream segregation. For example, under the analysis-by-synthesis framework [3,4], the speech mixture is first decomposed into features and features belonging to an auditory stream are selectively grouped together to form a representation of that stream in later processing stages. In other words, the temporal envelope of the sound mixture is extracted in the cochlear while the temporal envelope of an auditory stream is resynthesized in cortex [12,13,16]. Given the relatively low spatial resolution of MEG and the lack of structural MRI scans from the participants, here we did not investigate where anatomically the temporal envelope of the direct sound is resynthesized. Future studies, possibly requiring intracranial neural recordings from humans or animal neurophysiology, are required to analyze where the segregation between speech and echo emerges along the auditory pathway. Additionally, the streaming model assumes that the 2 streams are not just segregated but also encoded in different manners [39,50,51]. When the listener attends to one stream and ignores the other stream, many studies have demonstrated that the 2 streams are neurally segregated and differentially encoded depending on whether the stream is attended to [12,13,16,52]. Here, since the 2 streams differed by a time delay, it was possible that the leading stream suppressed the lagging stream through stream-level neural adaptation. For example, when a syllable was recognized from the leading stream, it could potentially adapt the neural response to the same syllable in the lagging stream, leading to differential encoding of the leading and lagging streams. This kind of adaptation occurred together with or after auditory stream segregation and was different from the neural adaptation that only considers the temporal envelope of the auditory stimulus.

The current experiment also suggested that, in an echoic environment, the auditory system could more reliably restore the very low-frequency components of the speech envelope compared with the higher-frequency components. In Experiment 1, the speech envelope was fully restored at 2 and 4 Hz but was only partially restored at 6 Hz (Fig 2B). Similarly, in Experiment 2, the speech envelope was fully restored at 2 Hz, partially restored at 4 Hz, and not restored at 6 Hz (Fig 3C). Similarly, in Experiment 3, evidence for auditory streaming was only observed below 5 Hz (S3 Fig). More reliable neural encoding of slower-than-faster speech envelopes had also been observed in previous studies. For example, neural tracking of speech envelope is more robust to a noisy listening environment at lower frequencies, e.g., near 1 Hz, than at higher frequencies, e.g., near 5 Hz [17]. Similarly, when the spectral resolution of speech is reduced through noise vocoding, neural tracking of speech envelope enhances at lower frequencies, e.g., near 1 Hz, but decreases at higher frequencies, e.g., near 5 Hz [53]. One possibility is that the auditory system needs to integrate over a longer time window to reliably extract speech information in a more challenging listening environment [54]. Here, in Experiments 1 and 2, the phase coherence below 2 Hz was generally enhanced for echoic speech than anechoic speech (Figs 2B, 3C, and S2), which might be explained by the increase in sound intensity for echoic speech compared with anechoic speech at low frequencies, as shown in the modulation spectrum (Fig 1B and 1C).

Finally, the MEG response could be better explained by the streaming model than the mixture model even when the listener's attention was diverted by a silent movie (Experiment 3). Movie watching can attract attention but does not impose high processing demand. Therefore, the current results suggested that the segregation of the direct sound and the echo occurred with minimal involvement of top-down attention, and further studies should investigate whether the result remains when participants are distracted by more demanding auditory tasks. In the current study, the predictive power of the streaming model did not significantly differ between the story-listening task and the movie-watching task, consistent with previous results that the envelope-tracking response is minimally modulated by cross-modal attention [55]. It has been heavily debated whether auditory stream segregation requires attention. Some theories and empirical evidence suggest that top-down attention is necessary to parse a complex auditory scene into auditory streams [3,56], while others argue that primitive auditory streaming can occur preattentively [11,57]. Auditory streaming is a complex process and it is possible that different modules involved in auditory streaming are differentially modulated by attention [16,58,59], perceptual similarity between streams [50], and task demand [51]. In the current experiment, the immediate repetition between direct sound and the echo served as an effective cue for speech segregation, and previous studies have indeed demonstrated that auditory streaming based on repetitions in the sound pattern could occur without the engagement of top-down attention [60].

In summary, the current study demonstrated that echo cancelation cannot be solved by neural adaptation mechanisms that can effectively cancel reverberation in typical daily acoustic environments. Instead, the processing of echoic speech involves speech segregation mechanisms that are more similar to the mechanisms engaged in processing speech in multi-talker environments. In other words, speech processing in an echoic environment is better viewed as an informational masking problem instead of an energetic masking problem [6].

## Materials and methods

### Participants

There are 51 participants (19 to 33 years old, mean age, 24.6 years) participated in the study in total, with 15 (5 male) participated in Experiment 1, 12 (3 male) participated in Experiment 2, and 24 (11 male) participated in Experiment 3. Three participants of Experiment 3 were excluded since the trigger signaling sound onset was missing. All participants were right-handed native Chinese speakers, with no self-reported hearing loss or neurological disorders. The experimental procedures were approved by the Ethics Committee of the College of Biomedical Engineering and Instrument, Zhejiang University (No. 2022–001). The participants provided written consent and were paid.

### Stimuli

The speech stimulus was taken from the beginning of a narration of the story Thatched Memories by Wenxuan Cao. Three segments of recording were used in the experiment, each of which lasted for about 13 min, with the constraint that the segment ended at the end of a sentence. The actual duration of the 3 segments was 12'59", 13'11", and 12'56", respectively. Each segment was used to generate the stimulus in 1 stimulus condition. Since each experiment consisted of 3 stimulus conditions, all 3 segments were presented in each experiment and their presentation order was kept the same, consistent with their order in the story. The order of stimulus conditions, i.e., how the speech was manipulated, however, was randomized and described in Procedures.

The echoic speech was constructed by delaying a speech signal by time $\tau$ and adding it back to the original signal (Fig 1A). If a speech signal was denoted as $s(t)$, echoic speech could be

expressed as $As(t) + s(t—\tau)$. The amplitude parameter $A$ was 1 in Experiments 1 and 3 and was 0.5 in Experiment 2. In the anechoic condition, only $s(t)$ was presented. When $A$ was 1, the echo had a maximal influence on the modulation spectrum and could completely notch out the speech envelope at echo-related frequencies. When $A$ was 0.5, the echo was more intensive than the direct sound and simulations showed that the neural adaptation slightly attenuated the neural response to the echo and rendered the neural responses to the echo and the direct sound of a more similar amplitude. Consequently, neural adaptation aggregated the influence of an echo on neural tracking of speech envelope at echo-related frequencies. In Experiments 1 and 3, the echoic speech was 3.01 dB stronger than the anechoic speech in terms of RMS of the sound waveform. In Experiment 2, the echoic speech was 0.97 dB stronger than the anechoic speech.

In Experiments 1 and 2, the delay $\tau$ was either 0.125 s or 0.25 s in a stimulus condition and remained unchanged throughout each approximately 13 min stimulus. We tested these 2 delays since a recent study demonstrated that a 0.125- and 0.25-s echo selectively attenuate the speech envelope at 4 Hz and 2 Hz [30], respectively, and these modulation frequencies were critical for speech intelligibility [23,26,27]. In Experiment 3, the echoic speech had a variable delay. The delay varied whenever there was a pause longer than 500 ms in speech, and each time the delay was independently drawn from a uniform distribution between 0.125 s and 0.25 s. In the current speech materials, the interval between paused longer than 500 ms was between 0.4 s and 11.2 s (3.4 s on average).

The 1-channel vocoded speech in Experiment 3 was constructed based on echoic speech with a variable delay. The direct sound and the echo were noise-vocoded respectively at first, and then were mixed to form the echoic vocoded speech. The speech envelope was extracted using the Hilbert Transform and was low-pass filtered below 160 Hz (fourth-order Butterworth filter). The 1-channel vocoded speech was generated by modulating Gaussian white noise (filtered below 4 kHz) with the temporal envelope of echoic speech. The RMS intensity of the noise-vocoded stimulus was adjusted to match that of echoic speech without noise-vocoding.

## Procedures

Participants sat in a quiet room while listening to speech, and their neural responses were recorded using MEG. All speech stimuli were diotically presented through headphones. Each experiment lasted for about 50 min.

**Experiment 1.** Experiment 1 consisted of 3 conditions that separately presented anechoic speech, echoic speech with a constant 0.125-s delay, and echoic speech with a constant 0.25-s delay. Each condition was presented in a separate block, and the order of the conditions was randomized among participants. While listening to the speech, the participants fixated on the cross to reduce eye movements. The participants were instructed to attend to speech and answer 2 comprehension questions after each block.

**Experiment 2.** The procedure of Experiment 2 was identical to the procedure in Experiment 1, except for a change in the power ratio between echo and direct sound.

**Experiment 3.** Experiment 3 also consisted of 3 conditions, which presented echoic speech with a variable delay and echoic vocoded speech. The 3 conditions were presented in 2 blocks, of which the order was randomized across participants. The first condition was presented in 1 block and the other 2 conditions were presented in the second block. The first condition presented echoic speech, and the participants performed the same question-answering task in Experiments 1 and 2. The second and third conditions presented the echoic speech and the echoic vocoded speech, respectively. The 2 conditions were presented in the same block but their order was randomized among participants. In the block, the participants were asked

to attend to a silent movie with subtitles (The Little Prince) and ignore any sound. The movie and subtitles could also generate neural responses but these responses were uncorrelated with the speech stimulus presented in the experiment. Also, the processing of subtitles might engage neural pathways that partially overlap with the processing of speech. The speech was presented about 5 min after the onset of the movie, to make sure that participants were already engaged in the movie-watching task. No break was made between the 2 conditions with the same block and the movie continued throughout. After the experiment, the participants had to answer simple comprehension questions about the movies and whether they noticed any auditory stimulus during the block. All participants answered the comprehension questions correctly.

## Data acquisition and preprocessing

Magnetoencephalographic (MEG) responses were recorded through a 306-sensor whole-head MEG system (MEGIN TRIUXTM neo, Finland) at the Institute of Neuroscience, Chinese Academy of Sciences. The system had 102 magnetometers and 204 planar gradiometers, and the MEG signals were sampled at 1 kHz. All data preprocessing and analysis were performed using MATLAB (The MathWorks, Natick, Massachusetts, United States of America). The temporal signal space separation (tSSS) was used to remove the external interference from the MEG responses [61]. Afterwards, the MEG signals were down-sampled to 100 Hz, and band-pass filtered between 0.8 and 10 Hz using a linear-phase finite impulse response (FIR) filter (6-s Hamming window, 6 dB attenuation at the cut-off frequencies). The linear-phase FIR filter caused a 3-s constant time delay to the input, which was compensated by removing the first 3 s samples in the filter output.

## Data analysis

**Modulation spectrum.** The modulation spectrum, i.e., the spectrum of the temporal envelope, was computed following the procedure in Ding and colleagues. In brief, a cochlear model was applied to extract the temporal envelope of speech in 128 narrow frequency bands [62], and each narrowband envelope was transformed into the frequency domain using the discrete Fourier transformation (DFT). The modulation spectrum summed the DFT spectra in individual frequency bands, and its maximal amplitude was normalized by the maximal amplitude of the modulation spectrum of anechoic speech.

**Phase coherence spectrum.** The phase coherence spectrum characterized the phase locking between the stimulus envelope and neural response, in each frequency band. The sound envelope was extracted by applying full-wave rectification to the sound, and low-pass filtering the signal below 50 Hz. The sound envelope and the neural response were both segmented into non-overlapping 2 s time epochs, and each epoch was transformed into the frequency domain using the DFT. For each complex-valued DFT coefficient, the phase angle was extracted. The phase for stimulus envelope and neural response were denoted as $\alpha_{ft}$ and $\beta_{ft}$, respectively, for frequency $f$ and epoch $t$, and the phase difference between response and stimulus was therefore $\theta_{ft} = \alpha_{ft} - \beta_{ft}$. The phase coherence spectrum was defined as follows:

$$C(f) = \frac{\left(\sum_{t=1}^{T} \cos(\theta_{ft})\right)^2 + \left(\sum_{t=1}^{T} \sin(\theta_{ft})\right)^2}{T^2}, \tag{1}$$

where $C(f)$ was the phase coherence at frequency $f$, and $T$ is the total number of epochs. The phase coherence was between 0 and 1. High phase coherence indicated that the phase difference between stimulus and response was consistent across epochs, i.e., stronger stimulus-response phase synchronization. The phase coherence spectrum was independently calculated for each MEG gradiometer. In this study, for anechoic speech, we calculated the phase coherence between the MEG response and the speech envelope. For echoic speech, we calculated the

phase coherence between the MEG response and the envelope of the direct sound, to characterize whether neural tracking of speech was influenced by the echo. The same result could be obtained when calculating the phase coherence between the MEG response and the envelope of the echo since there is a constant phase lag between the envelope of direct sound and the envelope of echo over time at each frequency.

**Simulation of a neural adaptation model.**   We simulated how synaptic depression and gain control influenced envelope-tracking neural response. We adopted the model used by Mesgarani and colleagues, which could effectively reduce the influence of reverberation on speech responses [9]. The model input, i.e., presynaptic activity, was the auditory spectrogram, which contained 128 frequency channels and could be viewed as a simulation of subcortical auditory responses [62]. The auditory spectrogram was sampled at 200 Hz, and the temporal envelope in each frequency channel, referred to as $s(t)$, was independently processed by the depression model. The neural response processed through the combined synaptic depression and the gain control model was:

$$r(t, c) = \max(s(t,c) - D(t,c), 0)G(t,c), \tag{2}$$

$$D(t, c) = V(1 + \sum_{n=0}^{\tau_{sd}} s(t-n, c)W(n)), \tag{3}$$

$$G(t, c) = \frac{1}{1 + \sum_{n=0}^{\tau_{gc}} r(t-n, c)W(n)}, \tag{4}$$

where $r(t, c)$ was the model output, i.e., the simulated postsynaptic response. $W(n)$ was a Hann window and the width equaled $\tau_{sd}$ and $\tau_{gc}$ in $D(t, c)$ and $G(t, c)$, respectively. The simulated envelope-tracking response was obtained by summing up the model output across all frequency channels. The model contained 3 parameters, i.e., $V$, $\tau_{sd}$, and $\tau_{gc}$, which were the firing threshold ($0 < V < 1$) and the time constants for the synaptic depression and the gain control ($0 < \tau_{sd} < 500$ ms, $0 < \tau_{gc} < 500$ ms). We tested possible combinations of the parameters (step size = 0.05, 50 ms, and 50 ms, for $V$, $\tau_{sd}$, and $\tau_{gc}$) and chose the combination to maximize the correlation coefficient between the simulated neural response and the envelope of direct sound that was averaged over the 4 echoic conditions (i.e., 2 delays × 2 experiments). Another synaptic depression model [63] and an optimal adaptive filter model [33] were also simulated but they were less effective at restoring the speech envelope at echo-related frequencies (S1 Fig).

**Temporal response function.**   The TRF was employed to model the time-domain relationship between the sound envelope and the MEG response. The mixture model assumed that the brain encoded the envelope of echoic speech and the model was described as:

$$r(t) = \sum_{\tau=1}^{D} \text{TRF}(\tau)s(t - \tau) + e(t), \tag{5}$$

where $r(t)$, $s(t)$, and $e(t)$ denoted the MEG response, the envelope of echoic speech, and the residual error, respectively. For both the mixture model and the 2 models described in the following, the length of the time integration window $D$ was 100, corresponding to 1 s. $\text{TRF}(t)$ was the TRF function, which could be interpreted as the response triggered by a unit power increase of the stimulus [17].

A streaming model assumed that the brain could segregate the echo and the direct sound into 2 streams and encode the 2 streams in different manners. The model was formulated as

follows:

$$r(t) = \sum_{\tau=1}^{D} \text{TRF}_d(\tau)s_d(t-\tau) + \sum_{\tau=1}^{D} \text{TRF}_e(\tau)s_e(t-\tau) + e(t), \tag{6}$$

where $s_d(t)$ and $s_e(t)$ were the envelopes of direct sound and echo, respectively, and $\text{TRF}_d(t)$ and $\text{TRF}_e(t)$ were the TRFs for the direct sound and the echo. In other words, $r(t)$ was modeled based on $[s_d(t-1), s_d(t-2),\ldots, s_d(t-D), s_e(t-1), s_e(t-2),\ldots, s_e(t-D)]$ in the streaming model, but was modeled based on just $[s_d(t-1), s_d(t-2),\ldots, s_d(t-D)]$ in the mixture model. If the 2 input signals, i.e., $s_d(t)$ and $s_e(t)$, were not fully correlated, the streaming model differed from the mixture model in terms of their inputs. Otherwise, e.g., if $s_d(t) = s_e(t—d)$, the input to the streaming model became $[s_d(t-1), s_d(t-2),\ldots, s_d(t-D-d)]$, which reduced to a mixture model.

An ideal model assumed that the brain only responded to the direct sound and the model was formulated as follows:

$$r(t) = \sum_{\tau=1}^{D} \text{TRF}_d(\tau)s_d(t-\tau) + e(t). \tag{7}$$

The TRF was independently computed for each model and each participant using ridge regression [64]. The predictive power of a model was defined as the correlation between the actual MEG response and the TRF prediction. The model was evaluated using 10-fold cross-validation. Specifically, each participant's MEG response was evenly divided into 10 segments. Nine segments were used to train the model, and the remaining segment was used to evaluate the predictive power of the model. The 10-fold cross-validation procedure resulted in 10 estimates of averaged predictive power and TRF. The regularization parameter for ridge regression was separately optimized for each model and each experimental condition.

**TRF with adapted input.** The TRF model could be adapted to consider the effect of neural adaptation, by concatenating a neural adaptation model with the TRF model [9]. The parameters and the estimation procedures of the TRF were the same as those described in the previous section. Parameters of the neural adaptation model, i.e., $V$, $\tau_{sd}$, $\tau_{gc}$, were separately optimized for each model and each experimental condition to maximize predictive power ($0 < V < 1$, $0 < \tau_{sd} < 800$ ms, $0 < \tau_{gc} < 1400$ ms; step size = 0.05, 200 ms, and 200 ms, for $V$, $\tau_{sd}$, and $\tau_{gc}$).

## Statistical analysis

To evaluate whether the phase coherence spectrum between stimulus and neural response at a frequency was significantly higher than the chance, the chance-level phase coherence spectrum was estimated using the following method [37,65]. After the envelope of stimulus and neural response were segmented into 2-s time bins, we shuffled all time bins for the stimulus envelope so that the envelope and response were randomly paired. We calculated the coherence of the phase difference between randomly paired neural response and stimulus envelope. This procedure was repeated 5,000 times, creating 5,000 chance-level phase coherence values. Then, the phase coherence values were averaged across channels and participants, for both the actual phase coherence and 5,000 chance-level phase coherence values. The significance level of the phase coherence at a frequency was $(M + 1)/5,001$ if it was lower than $M$ out of the 5,000 chance-level phase coherence values at that frequency (one-sided comparison).

When comparing the phase coherence or the TRF predictive power across conditions, we performed a permutation test, in which the condition label was switched for a subset of participants before calculating the mean difference across conditions. To obtain the null distribution

of the mean difference across conditions, the full set of $2^N$ possible permutations of $N$ participants was considered. Here, we applied a one-sided test to evaluate, e.g., whether the phase coherence is higher when listening to the anechoic speech than the echoic speech, and whether the streaming model has a higher predictive power than the alternative models. If the actual difference across conditions (mean over participants) was smaller than $M$ out of the $2^N$ permutations, the significant level was $(M + 1)/(2^N + 1)$. When multiple comparisons were performed, the $p$-value was adjusted using the FDR correction [66].

When comparing the phase coherence between Experiment 1 and Experiment 2, we performed a bias-corrected and accelerated bootstrap [67]. We applied a one-sided unpaired test to evaluate whether the phase coherence at an echo-related frequency was higher in Experiment 1 than in Experiment 2 using the following procedure: All participants in Experiment 1 were resampled 10,000 times with replacement, and the phase coherence for each set of resampled participants was averaged, creating 10,000 resampled mean values. If the mean phase coherence in Experiment 2 was greater than $M$ out of the 10,000 resampled mean values of Experiment 1, the significance level was $(M+1)/10,001$.

## Supporting information

**S1 Text. Supporting methods.** Simulations of a depression model and an optimal filter model.
(DOCX)

**S1 Fig. Phase coherence spectrum between the simulated neural response to echoic speech and the envelope of direct sound.** (A) Neural responses are simulated by processing the speech envelope using the synaptic depression model in David and colleagues (2009). (B) Neural responses are simulated by processing the speech envelope using the optimal adaptive filter similar to that of Ivanov and colleagues (2022). The underlying data can be found at https://zenodo.org/records/10472483.
(TIF)

**S2 Fig. Phase coherence at 1 Hz.** Each dot represents 1 individual and error bars represent 1 SEM. Dashed black lines show chance-level phase coherence. Phase coherence significantly higher than chance level and significant differences between conditions are marked (*
$p < 0.05$, ** $p < 0.01$, permutation test, FDR corrected). The underlying data can be found at https://zenodo.org/records/10472483.
(TIF)

**S3 Fig. TRF models in 0.8–5 Hz and 5–10 Hz frequency bands.** Predictive powers of the TRF models in 0.8–5 Hz (left plots) and 5–10 Hz (right plots), averaged over participants and MEG gradiometers. Gray dots show individual participants. Error bars represent 1 SEM across participants. The underlying data can be found at https://zenodo.org/records/10472483.
(TIF)

**S4 Fig. Illustration of 3 adapted TRF models.**
(TIF)

**S5 Fig. Predictive power difference between adapted TRF and TRF models.** The difference between the predictive powers of adapted TRF and TRF model, averaged over participants and MEG gradiometers. Gray dots show individual participants. Error bars represent 1 SEM across participants. The underlying data can be found at https://zenodo.org/records/10472483.
(TIF)

**S1 Audio. Exemplary echoic speeches for Experiment 1.**
(ZIP)

**S2 Audio. Exemplary echoic speeches for Experiment 2.**
(ZIP)

**S3 Audio. Exemplary echoic speeches for Experiment 3.**
(ZIP)

## Acknowledgments

We thank the MEG facility of CAS CEBSIT/ION for the assistance in data collection, Wenyuan Yu and Jiajie Zou for constructive comments on an earlier version of the manuscript, and Lingjun Jin for proofreading the manuscript.

## Author Contributions

**Conceptualization:** Nai Ding.

**Formal analysis:** Jiaxin Gao.

**Funding acquisition:** Nai Ding.

**Investigation:** Mingxuan Fang, Nai Ding.

**Methodology:** Jiaxin Gao, Nai Ding.

**Supervision:** Nai Ding.

**Visualization:** Jiaxin Gao, Nai Ding.

**Writing – original draft:** Jiaxin Gao, Honghua Chen, Nai Ding.

**Writing – review & editing:** Jiaxin Gao, Honghua Chen, Nai Ding.

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
