## [Editor Report · Decision Letter 0]

20 Jul 2023

Dear Dr Ding, 

Thank you for submitting your manuscript entitled "Automatic Auditory Streaming Restores Missing Temporal Modulations in Echoic Speech" for consideration as a Research Article by PLOS Biology.

Your manuscript has now been evaluated by the PLOS Biology editorial staff as well as by an academic editor with relevant expertise and I am writing to let you know that we would like to send your submission out for external peer review.

Once your full submission is complete, your paper will undergo a series of checks in preparation for peer review. After your manuscript has passed the checks it will be sent out for review. To provide the metadata for your submission, please Login to Editorial Manager (https://www.editorialmanager.com/pbiology) within two working days, i.e. by Jul 22 2023 11:59PM.

Kind regards,

Christian

Christian Schnell, PhD

Senior Editor

PLOS Biology

cschnell@plos.org

---

## [Decision Letter · Decision Letter 1]

8 Sep 2023

Dear Dr Ding,

Thank you for your patience while your manuscript "Automatic Auditory Streaming Restores Missing Temporal Modulations in Echoic Speech" was peer-reviewed at PLOS Biology. It has now been evaluated by the PLOS Biology editors, an Academic Editor with relevant expertise, and by several independent reviewers. 

In light of the reviews, which you will find at the end of this email, we would like to invite you to revise the work to thoroughly address the reviewers' reports.

As you will see below, the reviewers find your study potentially interesting but share concerns about the presentation of the hypotheses, data, and analyses, in addition to some technical and statistical concerns.

Given the extent of revision needed, we cannot make a decision about publication until we have seen the revised manuscript and your response to the reviewers' comments. Your revised manuscript is likely to be sent for further evaluation by all or a subset of the reviewers.

**IMPORTANT - SUBMITTING YOUR REVISION**

*Re-submission Checklist*

*Published Peer Review*

*PLOS Data Policy*

*Blot and Gel Data Policy*

Sincerely,

Christian

Christian Schnell, PhD

Senior Editor

PLOS Biology

cschnell@plos.org

REVIEWS:

Reviewer #1: This study examines neural (MEG) responses to echoic speech, i.e. speech that is mixed with a delayed version of itself. It is inspired by the fact that we can understand such speech quite well as evidenced by the current era of zoom calls as well as previous studies from the same group. This observation implies that original speech is perceptually segregated from its echo.

The authors claim to show that, when exposed to echoic speech, MEG responses track the slow temporal modulations (< 10 Hz) that were present in original speech but reduced in the stimulus by the addition of the echo. They also show results from temporal response function (TRF) models that evidence a neural segregation into original speech and echo.

I am sympathetic with the authors' approach and find the topic interesting and certainly relevant for research on auditory scene analysis. However I am currently not convinced that their results support several of the main claims (including that in the title). This might be partly due to me not fully understanding the analytical approach and I would be grateful for some clarifications.

This mostly concerns what is shown in Fig. 1C,D and the corresponding results in Fig. 2. If I understand the rationale correctly, the envelope of the original speech includes power around 4 Hz (?) which is reduced by the addition of the echo. If the brain only tracked the incoming signal, then tracking at 4 Hz should therefore be reduced in the echoic speech condition. However if it segregates original speech and echo, then we should find 4 Hz tracking even in the echoic condition, as a reflection of stream segregation. If my summary is incorrect, then I would be grateful for more details on the study's aim. If it is correct, I do not understand why the authors do not show the envelope spectrum of the original speech, extract the dominant frequency (such as 4 Hz), and then test whether tracking of this frequency (e.g., phase coherence between MEG and speech envelope) is preserved in the echoic condition. An envelope spectrum for the echoic speech would also be helpful. If there is a peak in the tracking response to echoic speech that corresponds to one in the envelope spectrum for original but not echoic speech, I would be convinced that the temporal modulation is restored as claimed.

I agree that conventional analyses of speech tracking are often based on phase-locking between neural signal and speech, but it is not fully clear to me how this is linked to the phase coherence between original and echoic speech, and the focus on notches. If the brain does not segregate original speech and echo, I would still expect to see reduced tracking as a result from reduced power of temporal modulations in the echoic stimulus. Similarly, the term "echo-related frequencies" was not clear to me (why are they echo related? Is there power in the echo at those frequencies?). 

Other, including minor comments (random order):

- I wonder whether the term "automatic" is really necessary as this is hard to demonstrate.

- At some point the nomenclature changes from "direct speech" to "anechoic speech", this was a bit confusing and it would be good to clarify that both refer to the same stimulus. 

- If the aim is to compare tracking at certain frequencies across conditions, it might be important to correct for differences in 1/f slopes that are visible in Fig. 2.

Reviewer #2: The ms. tests the possible mechanisms of the previously observed excellent comprehension of echoed speech. The results of two experiments show that adaptation models do not accurately explain the correspondence between magnetic brain activity and the speech envelope. Modeling the results of the third experiment shows that a general linear model based on the assumption that the original speech and its echo are segregated and separately processed in the brain better predict the observed magnetic brain activity than similar models based either on the assumption that the full signal (original speech + echo) or only the original speech signal is processed as the input of the model. The paper thus suggests that in contrast to how reverberation is handled by the brain, echoes are processed with the help of first segregating them from the original speech signal.

This is an excellent ms., using computational modeling to decide about the likely algorithms the brain uses to extract meaning from echoed speech. The study is well-motivated, and the relevant literature is cited. The experiments are mostly well designed and address the important aspects of the questions of the ms.; measurements and signal processing are clear and to the point (see some minor comments, though). Thus, the results are reliable and support the conclusions of the ms.

The following comments are meant to improve the readability and impact of the ms.

Major point:

The study consists of two parts. The first part shows the problems adaptation models encounter in explaining the correspondence between the signal of the echoed speech segment and the envelope of the concurrent magnetic brain activity (Experiments I and II). The second part then tests whether general linear models better explain the observed brain activity if they assume that the original speech and its echo have been segregated prior to entering the modeled processing compared to entering the composite or the original (unechoed) signal.

There are some issues with the second part, which are omitted or not discussed in the ms.

A) It is implicitly assumed that auditory stream segregation precedes the processes resulting in speech envelope tracking. This is quite likely not the case. The authors themselves argue that auditory stream segregation is helped by low level cues (which include the dynamic tracking of speech energy) as well as high-level ones (which include comprehending the speech stream so far). This makes it very likely that speech processing (including echo processing) goes hand-in-hand with auditory stream segregation. Second, it is quite unlikely that the output of auditory stream segregation is a restored sound signal, which could be fed into to further processing (including the processing of echoes). Thus, the related point needs to be made much more carefully (including the abstract and the conclusions), and limitations of this approach should be spelled out.

B) The paper is aimed to make the point that assuming segregation of the echo from the original speech is a more likely approach of the brain to the problem of echoes than neural adaptation. However, the two are never directly compared. Thus, it should be discussed how well the models compared in the second part cover adaptation or test how well adaptation models would work if stream segregation preceded the adaptation phenomena. In fact, adaptation might not be directly comparable to stream segregation, because while stream segregation is a high-level process, adaptation is often treated as a low-level mechanism. (Or spell out a high-level version of adaptation to be contrasted with stream segregation, if this is how you wish to consider adaptation.) Thus, it is quite possible that there is no real contradiction between the two, as they represent different levels of explanation. These issues should be carefully discussed, else the two parts of the ms. fall apart.

Minor points:

ll. 131-132: "For speech with a 0.125-s echo, the phase coherence spectrum showed notches at 4 and 12 Hz." Phase coherence is between two signals. The sentence does not tell what was compared with what for the 0.125 echo.

II. 245-247: This is the first instance of the reader encountering the term "baseline model", because the methods are described at the end of the ms. A more detailed explanation is needed, telling what is meant by the baseline model (my "Major point" above should also be considered).

Figures 5A and 6A are not very informative as the corresponding information is clearly provided in the main text.

ll. 455-464: "s(t)" and "As(t)+s(t-τ)" carry different energy. Why were the RMS intensities not matched? The possible effects of different energies should be mentioned.

ll. 476-481: The description gives the impression that vocoding was applied to the composite (original+echo) signal. If this was the case, then it should be shown that the vocoded signal is still an echoed sound signal. Otherwise, this condition does not test the echo phenomenon. To model how echoes can be resolved for a vocoded signal, the order should be vocoding the original signal than apply echoing.

II. 509-510: Using a subtitled movie for the passive experiments may have caused interference between the texts read (and possibly subvocalized) and the test sounds. This should be noted in the text.

Reviewer #3: This paper investigates the robustness of human speech perception to the presence of echoes, as occur in some natural environments and as are common in online videoconferencing. Such echoes can distort the amplitude envelope of a speech signal entering the ear, but have little effect on human perception. First, the authors provide evidence that human robustness cannot be explained by adaptation mechanisms previously proposed to explain robustness to reverberation. Second, they show that the envelope of the "direct" speech signal (i.e., without the echo) is represented in the auditory cortex more than would be expected without some compensatory mechanism, as evidenced by phase coherence between the MEG signal and the direct speech envelope. Third, they show that the MEG response to echoic speech is best explained by a TRF model that has two TRFs, one for the direct speech envelope and one for the echo envelope (this does better than a single TRF fit to the envelope of the combined direct+echo speech, or to just the direct speech). This latter result also held for echoic speech in which the echo delay varied over time, and when listeners were watching a movie rather than performing a task with the speech, but was eliminated when the speech was noise-vocoded (a manipulation intended to prevent the direct and echo speech from being streamed). The authors conclude that robustness to echos of this sort is mediated by streaming, with the echo represented as distinct from the direct speech. 

My overall assessment is that this is good and novel work. I think the conclusions are supported by the data, and my main suggestions regard the clarity of presentation, which I think can be improved. I am supportive of publication following revisions to address these issues.

1. Issues of interpretation that merit further discussion:

a) I think the authors should make it clear that the "streaming model" that they advocate is a bit different from "generic" streaming (as would happen if someone were listening to two distinct and independent voices). The TRF model they use would not work, I think, if applied to this more generic context. In other words, there is something different about the way the echo is encoded compared to the direct sound, which is why it is possible to use the dual-TRF model in this case. It is perhaps analogous to the situation where one stream is attended and another is not.

b) The results could be taken as evidence that the neural tracking of the speech envelope is better understood as a consequence of speech segregation rather than a cause. Please discuss.

2. Issues that need to be better explained to the reader:

a) Throughout the paper, it would help to provide more guidance re: the interpretation of results. I would have appreciated being given an explicit conclusion at the end of most paragraphs, confirming my guess as to what the results mean. I have noted a few specific places below. Pages 9 and 10 were particularly tough going - they seemed like a long list of results, with the burden being left on the reader to figure out how they fit together.

b) I think you should help the reader think through the consequence of the echo and direct sound being correlated. Please spell out the argument here in more detail, up front. This eventually becomes somewhat clear with the variable delays in Experiment 3, but I think should be addressed head on earlier in the paper. It is natural for the reader to wonder what happens if you compute phase coherence with the echo rather than the direct sound. This is not a meaningful distinction in Experiments 1 and 2, but you should explain that up front, and also explain why it is not a problem for the interpretation, but that it leaves some issues open. The same issue is present for the TRF model being fit to Experiments 1 and 2.

Specific Comments:

The English could use proofreading for grammar (in particular, the use of "the", which seemed to be missing in many places where I would have expected it).

100: phase coherence needs to be explained briefly in the main text for a general audience

102: unclear - I presume results are averaged across 270 examples, one for each of the 270 IRs? Was the same speech envelope used in each case?

108: please briefly specified how the parameters were optimized

132-133: explain for the non-expert why the notches occur where they do

151: please spell out the conclusion, rather than leaving it implicit

164: "comprehensions" is the wrong word

172: please state the conclusion we should draw from this set of results

195-198: Is this result predicted by some hypothesis? How are we to interpret it? Please tell me what I should think about it.

218: please state the conclusion we should draw from this set of results

Figures 3 and 4 should be reversed in order, in my opinion, to create parallel organization of the results of experiments 1 and 2.

264: please spell out the logic of the argument for why the TRF analysis provides evidence for segregation into separate streams

270: the word "barely" was confusing to me.

273: same here

305: please give the conclusion the reader should draw from these results

349: I think it would make sense to acknowledge that such echoes do sometimes occur in natural environments, and that this could ultimately be the reason why we are able to handle echoes in this way. It at least seems like a plausible possibility. I don't think the results are exclusively relevant to Zoom…

366-368: this idea should be explained earlier, when you show the initial modeling results

394: reference 34 should be mentioned in the introduction

Reviewer #4: In this article by Gao et al. the neural tracking of speech envelope in the presence of a single, long-delay echo is investigated using magnetoencephalography. Potentially, this article may be an interesting addition to the growing number of studies that investigated continuous speech streams using some form of MEG, EEG, ECoG-based tracking. The article put forward the hypothesis that, in the presence of a single echo, the direct speech and the echo are automatically segregated in two streams in the auditory cortex. This is a suggestive and interesting hypothesis, however, I am not convinced that the evidence provided is sufficiently strong to convincingly support it. As described below in detail, I find that 1) not all results are actually consistent with this hypothesis, 2) currently unexplained effects may be more relevant than those examined and that 3) the statistical approach can be improved and should be more conservative. Below are my specific comments:

The first part of the results section (Figure 1A, 1B , 1C) describes an analysis of the stimuli showing that the echo creates phase distortions to the speech envelope at specific temporal modulation frequencies (related to the echo delay). Based on this observation, the authors postulate their working hypothesis that successful neural tracking of missing echo-related components provides evidence for auditory stream segregation of the directed and shifted speech stream. 

I find some of the results in Figure 1 unintuitive: In Figure 1C, for instance, phase distortion is larger when the echo has the same amplitude as the direct sound (Exp. 1) than when the echo is twice stronger than the direct sound (Exp.2). What is the reason for this? May this reflect somehow the overall energy/loudness of the stimuli? Were anechoic and echoic speech normalized in energy/loudness? How? (In Figure 1D, the pattern of results for the neural adaptation simulation is more like one would expect, with stronger effects for the more intense echo signal).

This first part also includes an additional analysis conducted to exclude the possibility that neural adaptation mechanisms in the auditory system would compensate these echo-related distortions, the authors simulated (with several models) the neural responses with adaptation mechanisms to the echoic (and anechoic) speech and calculated their coherence them with the anechoic speech envelope. Results showed a reduced phase coherence between simulated neural responses and echoic speech envelope at the echo-related modulation frequencies. These results are taken as evidence that neural adaptation mechanisms are not sufficient to restore phase coherence in the presence of echoes and an alternative mechanism is required.

While I appreciate the potential relevance of this analysis and the need to exclude obvious alternative hypotheses, I am not sure that this is the right place for this description. Without first reading the MEG results, I missed the message this analysis wants to convey. However, it became clearer after a second reading of the paper. I would consider moving them later in the paper or, alternatively, in the supplementary materials.

Figure 2 shows the results of the MEG tracking of anechoic and echoic (Exp.1, 0 dB, 0.125 s and 0.25 s) speech on the phase coherence at different temporal modulations. Here the authors focus on the lack of differences between the tracking of echoic and anechoic speech at the echo-related modulation frequencies support to their hypothesis with ad hoc statistical tests (Figure 3, see below my comments on the statistical analysis). These tests confirmed the hypothesis at 4 Hz for 0.125 s delay, and at 2 Hz for 0.25 s, but not at 6 Hz for 0.25 s. 

I was surprised that the authors give little (or no) attention to what seems to be a major effect of the echo in Figure 2: an enhanced phase coherence at the low modulation frequencies. What is the interpretation of this effect? I feel that if this point is not addressed convincingly, it becomes difficult shifting the readers' focus on the much smaller echo-related effects. This is not addressed in the Discussion either. 

My reading of the results of Experiment 2 (in Figure 4) is that they are only partially consistent with those of Experiment 1 and with the hypothesized mechanisms. In two out of the three relevant tests (i.e. 4Hz for 0.125 and 6 Hz for 0.25s) there is significant reduction of phase locking compared with the responses to the anechoic speech. Additionally, I am not convinced of the robustness of the performed statistical analyses. If I understood correctly the actual implementation, the bootstrap procedure implemented to perform the two-sided paired comparison may be quite liberal, corresponding to a fixed-effects comparison. The authors could perform a random-effects non-parametric test by comparing the observed group average difference to a null distribution obtained by re-computing the group average difference after swapping the condition for a randomly selected subset of subjects (2^Nsubjects number of permutations possible). 

Overall, for what concerns the analysis of modulation-specific phase coherence, I do not find that the evidence provided to support the formulated hypothesis is compelling, both because of inconsistencies in the observed patterns of results and because there seem to be other bigger effects that may be more relevant than those examined. In addition, I thinks that the statistical approach needs to be clarified and possibly revised.

To further support the hypothesis that measured MEG responses reflect the auditory streaming of direct sound and echo the authors conducted a TRF analyses, comparing a baseline (echoic speech), streaming (direct sound, echo) and idealized (direct sound) models. In line with the streaming hypothesis, results reported in Figure 5 seem to indicate that the two predictors model predict is (slightly) better than the other models. Results from Experiment 3 (Figure 6) with variable delays confirm that the streaming model has a slight higher predictive power except in the case where relevant speech cues have been removed (vocoded speech). In principle, this is, in my view, the strongest evidence the authors in support of the streaming hypothesis. However, the description of the methods employed does not allow judging whether the conclusions are sound. In particular, I am missing two aspects: 

- A precise description on how the data for model fitting and estimation of the predictive power is separated (e.g. at which level of the processing pipeline). This is important to assess to what extend fitting and testing data can be considered independent.

- A precise description on how model differences are assessed statistically. As a specific description is missing, I am assuming that this is done as in the phase coherence analysis. Thus, my previous comment on the need of implementing random-effects statistics applies also in this case. 

Finally, there is quite some discrepancy between the shape of the TRFs for Experiment 1 and 2, almost with a switch between signals. This may not be necessarily a problem, but I think it deserves to be discussed. In Experiment 3 TRFs are more consistent but also in this case it would be interesting to discuss the shape of the obtained TRFs.

---

## [Editor Report · Decision Letter 2]

19 Dec 2023

Dear Dr Ding,

Thank you for your patience while we considered your revised manuscript "Auditory Stream Segregation Restores Missing Temporal Modulations in Echoic Speech" for publication as a Research Article at PLOS Biology. This revised version of your manuscript has been evaluated by the PLOS Biology editors and the Academic Editor.

Based on our Academic Editor's assessment of your revision, we are likely to accept this manuscript for publication, provided you satisfactorily address the following data and other policy-related requests:

* We would like to suggest a different title to improve readability: "Original speech and its echo are segregated and separately processed in the human brain"

* Please add links to the funding agencies in the Financial Disclosure statement in the manuscript details

* Please provide a blurb which (if accepted) will be included in our weekly and monthly Electronic Table of Contents, sent out to readers of PLOS Biology, and may be used to promote your article in social media. The blurb should be about 30-40 words long and is subject to editorial changes. It should, without exaggeration, entice people to read your manuscript. It should not be redundant with the title and should not contain acronyms or abbreviations.

DATA POLICY:

Regardless of the method selected, please ensure that you provide the individual numerical values that underlie the summary data displayed in the following figure panels as they are essential for readers to assess your analysis and to reproduce it: 2C, 3D, 4B, 4C, 5A, 5B, 5C, 6B–6F, and similar figures in the supplementary information

* Thank you for depositing your data and code at https://github.com/Yee-Gao/E-Encoding. Please also assign a DOI so that the repository is citable and versioned for your paper. Zenodo is one of the available tools for this.

We expect to receive your revised manuscript within two weeks. 

*Published Peer Review History*

*Press*

Sincerely,

Christian

Christian Schnell, PhD

Senior Editor,

cschnell@plos.org,

PLOS Biology

---

## [Editor Report · Decision Letter 3]

8 Jan 2024

Dear Dr Ding,

Happy New Year!

Thank you for your patience while we considered your revised manuscript "Original Speech and Its Echo are Segregated and Separately Processed in the Human Brain" for publication as a Research Article at PLOS Biology. This revised version of your manuscript has been evaluated by the PLOS Biology editors.

Thank you for responding to the editorial requests which are now largely addressed. The only open point is the source data, i.e. the individual quantitative observations that underlie the data summarized in the figures and results of your paper.

Regardless of the method selected, please ensure that you provide the individual numerical values that underlie the summary data displayed in the following figure panels as they are essential for readers to assess your analysis and to reproduce it: 2C, 3D, 4B, 4C, 5A, 5B, 5C, 6B–6F, and similar figures in the supplementary information

Please also ensure that figure legends in your manuscript include information on where the underlying data can be found (for example: "Source data can be found in S1 Data."), and ensure your supplemental data file/s has a legend.

We expect to receive your revised manuscript within two weeks. 

*Published Peer Review History*

*Press*

Sincerely,

Christian

Christian Schnell, PhD

Senior Editor,

cschnell@plos.org,

PLOS Biology

---

## [Editor Report · Decision Letter 4]

15 Jan 2024

Dear Dr Ding,

Thank you for the submission of your revised Research Article "Original Speech and Its Echo are Segregated and Separately Processed in the Human Brain" for publication in PLOS Biology. On behalf of my colleagues and the Academic Editor, Manuel Malmierca, I am pleased to say that we can in principle accept your manuscript for publication, provided you address any remaining formatting and reporting issues. These will be detailed in an email you should receive within 2-3 business days from our colleagues in the journal operations team; no action is required from you until then. Please note that we will not be able to formally accept your manuscript and schedule it for publication until you have completed any requested changes.

PRESS

Sincerely, 

Christian

Christian Schnell, PhD, PhD

Senior Editor

PLOS Biology

cschnell@plos.org